# DYT1 dystonia increases risk taking in humans

**David Arkadir[1]\*, Angela Radulescu[2,3], Deborah Raymond[4], Naomi Lubarr[4], Susan B Bressman[4], Pietro Mazzoni[5], Yael Niv[2,3]\***

[1]Department of Neurology, Hadassah Medical Center and the Hebrew University, Jerusalem, Israel; [2]Department of Psychology, Princeton University, Princeton, United States; [3]Princeton Neuroscience Institute, Princeton University, Princeton, United States; [4]Department of Neurology, Beth Israel Medical Center, New York, United States; [5]The Neurological Institute, Columbia University, New York, United States

**Abstract** It has been difficult to link synaptic modification to overt behavioral changes. Rodent models of DYT1 dystonia, a motor disorder caused by a single gene mutation, demonstrate increased long-term potentiation and decreased long-term depression in corticostriatal synapses. Computationally, such asymmetric learning predicts risk taking in probabilistic tasks. Here we demonstrate abnormal risk taking in DYT1 dystonia patients, which is correlated with disease severity, thereby supporting striatal plasticity in shaping choice behavior in humans.

\*For correspondence: arkadir@gmail.com (DA); yael@princeton.edu (YN)

## Introduction

DYT1 dystonia is a rare, dominantly inherited form of dystonia, caused almost exclusively by a specific deletion of three base pairs in the *TOR1A* gene (*Ozelius et al., 1997*). Clinically, DYT1 dystonia is characterized by variable severity of sustained or intermittent muscle contractions that produce abnormal movements. DYT1 dystonia patients have normal intelligence, and post-mortem examination of their brains does not reveal obvious abnormalities or evidence of neurodegeneration (*Paudel et al., 2012*). Nevertheless, research in two different rodent models of DYT1 dystonia points to the existence of a fundamental deficit in synaptic plasticity. Specifically, brain slices of transgenic rodents expressing the human mutant *TOR1A* gene show abnormally strong long-term potentiation (LTP; *Martella et al., 2009*) and weak, or even absent, long-term depression (LTD; *Grundmann et al., 2012*; *Martella et al., 2009*) in corticostriatal synapses, as compared to wild-type controls.

Reinforcement learning theory (*Sutton and Barto, 1998*) hypothesizes that dopamine-dependent synaptic plasticity in corticostriatal networks is the neuronal substrate for learning through trial and error (*Barnes et al., 2005*; *Barto, 1995*; *Schultz et al., 1997*). The core assumptions of this theory are that (1) dopamine release in the striatum signals errors in the prediction of reward, with dopamine levels increasing following successful actions (to signal a positive prediction error) and decreasing when actions fail to achieve the expected outcome (a negative prediction error), (2) fluctuations in dopamine modulate downstream plasticity in recently active corticostriatal synapses such that synapses responsible for positive prediction errors are strengthened through long-term potentiation (LTP), and those that led to disappointment are weakened through long-term depression (LTD) (*Reynolds et al., 2001*), and (3) the efficacy of corticostriatal transmission affects voluntary action selection.

Dopamine's role as a reinforcing signal for trial-and-error learning is supported by numerous findings (*Pessiglione et al., 2006*; *Schultz et al., 1997*; *Steinberg et al., 2013*), including in humans,

**eLife digest** We learn to choose better options and avoid worse ones through trial and error, but exactly how this happens is still unclear. One idea is that we learn 'values' for options: whenever we choose an option and get more reward than originally expected (for example, if an unappetizing-looking food turns out to be very tasty), the value of that option increases. Likewise, if we get less reward than expected, the chosen option's value decreases.

This learning process is hypothesized to work via the strengthening and weakening of connections between neurons in two parts of the brain: the cortex and the striatum. In this model, the activity of the neurons in the cortex represents the options, and the value of these options is represented by the activity of neurons in the striatum. Strengthening the connections is thought to increase the value of the stimulus, but this theory has been difficult to test.

In humans, a single genetic mutation causes a movement disorder called DYT1 dystonia, in which muscles contract involuntarily. In rodents, the same mutation causes the connections between the neurons in the cortex and the striatum to become too strong. If the theory about value learning is true, this strengthening should affect the decisions of patients that have DYT1 dystonia.

Arkadir et al. got healthy people and people with DYT1 dystonia to play a game where they had to choose between a 'sure' option and a 'risky' option. Picking the sure option guaranteed the player would receive a small amount of money, whereas the risky option gave either double this amount or nothing. The theory predicts that the double rewards should cause the patients to learn abnormally high values, which would lure them into making risky choices. Indeed, Arkadir et al. found that players with DYT1 dystonia were more likely to choose the risky option, with the people who had more severe symptoms of dystonia having a greater tendency towards taking risks.

Arkadir et al. showed that these results correspond with a model that suggests that people with DYT1 dystonia learn excessively from unexpected wins but show weakened learning after losses, causing them to over-estimate the value of risky choices. This imbalance mirrors the previous results that showed an inappropriate strengthening of the connections between neurons in rodents, and so suggests that similar changes occur in the brains of humans. Thus it appears that the changes in the strength of the connections between neurons translate into changes in behavior.

This pattern of results might also mean that the movement problems seen in people with DYT1 dystonia may be because they over-learn movements that previously led to a desired outcome and cannot sufficiently suppress movements that are no longer useful. Testing this idea will require further experiments.

where Parkinson's disease serves as a human model for altered dopaminergic transmission (*Frank et al., 2004*). However, the contribution of (dopamine modulated) corticostriatal plasticity to shaping action has remained unconfirmed in the behaving organism, as it is not clear that the behavioral effects of altered dopamine signaling in Parkinson's disease (and other conditions in which dopamine transmission is compromised) indeed stem from the role of dopamine in modulating plasticity. Towards this end, here we test whether DYT1 dystonia, where corticostriatal plasticity is suggested to be altered despite preserved dopaminergic signaling, leads to the behavioral effects predicted by reinforcement learning with imbalanced plasticity. In particular, our predictions stem from considering the effects of intact prediction errors on an altered plasticity mechanism that amplifies the effect of positive prediction errors (i.e., responds to positive prediction errors with more LTP than would otherwise occur in controls) and mutes the effects of negative prediction errors (that is, responds with weakened LTD as compared to controls).

We compared the behavior of DYT1 dystonia patients and healthy controls on an operant-learning paradigm with probabilistic rewards (*Niv et al., 2012*). Participants learned from trial and error to associate four different visual cues with monetary rewards (*Figure 1a*), optimizing their gain by selecting one of two cues in choice trials, and choosing the single available cue in forced trials. Three visual cues were associated with a payoff of 0¢, 5¢ and 10¢, respectively, while the fourth cue was associated with an unpredictable payoff of either 0¢ or 10¢ with equal probabilities (henceforth the 'risky 0/10¢' cue). Based on the findings in rodents with the DYT1 mutation, we predicted that

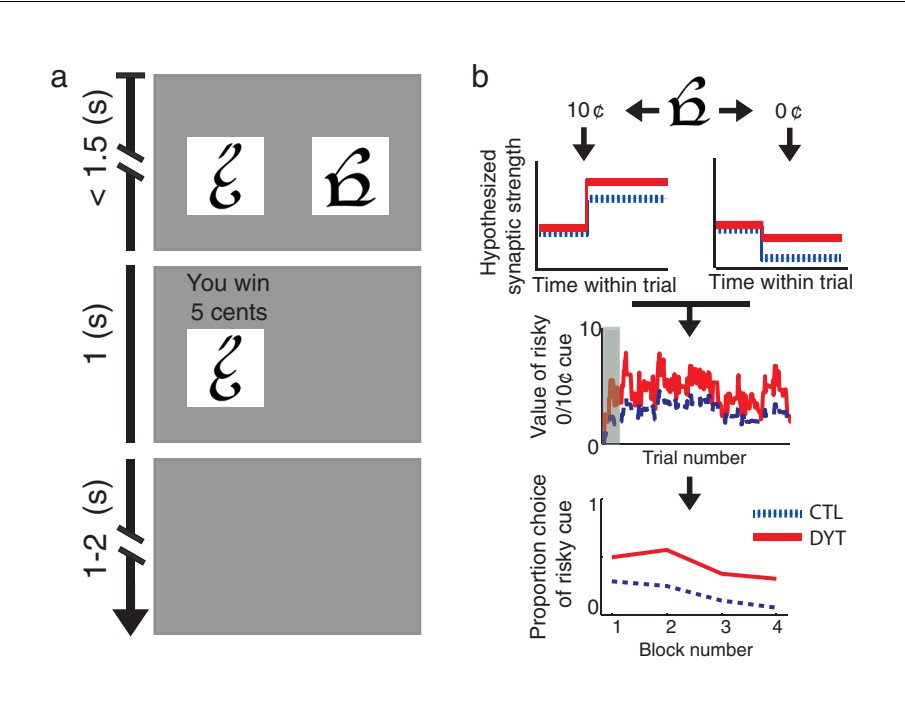

**Figure 1.** Behavioral task and hypothesis. (**a**) In 'choice trials' two visual cues were simultaneously presented on a computer screen. The participant was required to make a choice within 1.5 s. The chosen option and the outcome then appeared for 1 s, followed by a variable inter-trial interval. (**b**) Theoretical framework. Top: trials in which the risky cue is chosen and the obtained outcome is larger than expected (trials with a 10¢ outcome) should result in strengthening of corticostriatal connections (LTP), thereby increasing the expected value of the cue and the tendency to choose it in the future. Conversely, outcomes that are smaller than expected (0¢) should cause synaptic weakening (LTD) and a resulting decrease in choice probability. Middle: In DYT1 dystonia patients (red solid), increased LTP combined with decreased LTD are expected to result in an overall higher learned value for the risky cue, as compared to controls (blue dashed). In the model, this is reflected in higher probability of choosing the risky cue when presented together with sure 5¢ cue (Bottom). Simulations (1000 runs) used the actual order of trials and mean model parameters of each group as fit to participants' behavior. Gray shadow in the middle plot denotes trials in the initial training phase.

dystonia patients would learn preferentially from positive prediction errors (putatively due to abnormally strong LTP) and to a much lesser extent from negative prediction errors (due to weak LTD) (*Figure 1b*). As a result, they should show a stronger tendency to choose the risky cue as compared to healthy controls.

## Results

We tested 13 patients with DYT1 dystonia (8 women, 5 men, age 20–47, mean 28.6 years, henceforth DYT) and 13 healthy controls (CTL; 8 women, 5 men, age 19–46, mean 28.8 years), matched on an individual basis for sex and age (Mann-Whitney U test for age differences, $z = -0.59$, df = 24, $P = 0.55$), all with at least 13 years of education. Patients had no previous neurosurgical interventions for dystonia (including deep brain stimulation) and were tested before their scheduled dose of medication when possible (see Materials and methods). The number of aborted trials was similarly low in both groups (DYT $2.3 \pm 2.5$, CTL $1.1 \pm 1.2$, Mann-Whitney $z = -1.61$, df = 24, $P = 0.11$) and reactions times were well below the 1.5s response deadline (DYT 0.78s $\pm$ 0.11, CTL 0.71s $\pm$ 0.10, Mann-Whitney $z = -1.49$, df = 24, $P = 0.14$), confirming that motor symptoms of dystonia did not interfere with the minimal motor demands of the task.

Both groups quickly learned the task, and showed similarly high probabilities of choosing the best cue in trials in which a pair of sure cues (sure 0¢ vs. sure 5¢ or sure ¢5 vs. sure 10¢) appeared

together (mean probability correct choice: DYT1 0.92 ± 0.08, CTL 0.93 ± 0.05, Mann-Whitney z = 0.08, df = 24, P = 0.94; *Figure 2a*), as well as in trials in which the risky cue appeared together with either the sure 0¢ or sure 10¢ cues (mean probability correct: DYT1 0.84 ± 0.09, CTL 0.89 ± 0.04, Mann-Whitney z = −1.39, df = 24, P = 0.17; *Figure 2b*).

On trials in which the risky 0/10¢ cue appeared together with the equal-mean 5¢ sure cue, control participants showed risk-averse behavior, as is typically observed in such tasks (*Kahneman and Tversky, 1979*; *Niv et al., 2012*). In contrast, patients with DYT1 dystonia displayed significantly less risk aversion, choosing the risky stimulus more often than controls throughout the experiment (*Figure 3a*, Mann Whitney one-sided test for each block separately, all z > 1.68, df = 24, P < 0.05; Friedman's test for effect of group after correcting for the effect of time $\chi^2$ = 16.2, df = 1, P < 0.0001). Overall, the probability of choosing the risky cue was significantly higher among patients with dystonia than among healthy controls (*Figure 3b*, probability of choosing the risky cue over the sure cue DYT 0.44 ± 0.18, CTL 0.25 ± 0.20, Mann-Whitney z = 2.33, df = 24, P < 0.05).

To rule out the possibility that DYT1 patients were simply making choices randomly, causing their behavior to seem indifferent to risk, we divided all 0/10¢ versus 5¢ choice trials according to the outcome of the previous trial in which the risky 0/10¢ cue was chosen. As shown in *Figure 3c* (see also *Figure 3—figure supplement 1*), both groups chose the risky 0/10¢ cue significantly more often after a 10¢ 'win' than after a 0¢ 'loss' outcome (DYT P < 0.005, CTL P < 0.05, Wilcoxon signed-rank test), attesting to intact reinforcement learning in the DYT group (see *Figure 3—figure supplement 2*, for a reinforcement learning simulation of the same result). If anything, DYT1 dystonia patients showed a greater difference between trials following a win and those following a loss. We next tested for a correlation between risk-taking behavior and the clinical severity of dystonia, as rated on the day of the experiment (see Materials and methods). The results showed that patients with more

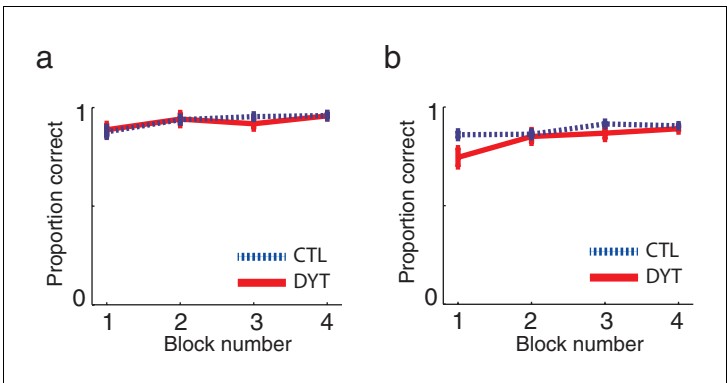

**Figure 2.** Learning curves did not differ between the groups. Mean probabilities (± s.e.m) of choosing the cue associated with the higher outcome, on average, (a) among pairs of two sure cues (15 trials per 'block') or (b) when the risky 0/10¢ cue was paired with a sure cue of 0¢ or 10¢ value (20 trials per 'block') confirmed that both groups quickly learned to choose the best cue in trials in which one cue was explicitly better than the other. These results verify that both groups understood the task instructions and could perform the task similarly well (in terms of choosing and executing their responses fast enough, etc.). Participants evidenced learning of values for deterministically-rewarded cues even in the first choice trials despite the fact that they were never informed verbally or otherwise of the monetary outcomes associated with each of the cues, and thus could only learn these from experience. However, for cues leading to deterministic outcomes, a little experience can go a long way (*Shteingart et al., 2013*), and participants received 16 training trials prior to the test phase. Our data suggest that learning in this phase did not differ between the groups: in the first 5 choice trials in the test phase that involved a pair of sure cues, the probability of a correct response was 0.78 ± 0.18 in the DYT group and 0.81 ± 0.07 in the CTL group (Mann-Whitney U test, df = 24, P = 0.59). We verified that that this level of performance could result from trial-and-error learning by simulating the behaviors of individuals using the best-fit learning rates (see Materials and methods). The simulation confirmed that both groups should show similar rates of success on the first 5 choice trials (DYT 0.81 ± 0.17 probability for correct choice, CTL 0.87 ± 0.13, Mann-Whitney U test z = 0.88, df = 24, P = 0.38) despite differences in learning rates from positive and negative prediction errors (see Results). Indeed the model, which started from initial values of 0 and learned only via reinforcement learning, performed on average *better* than participants.

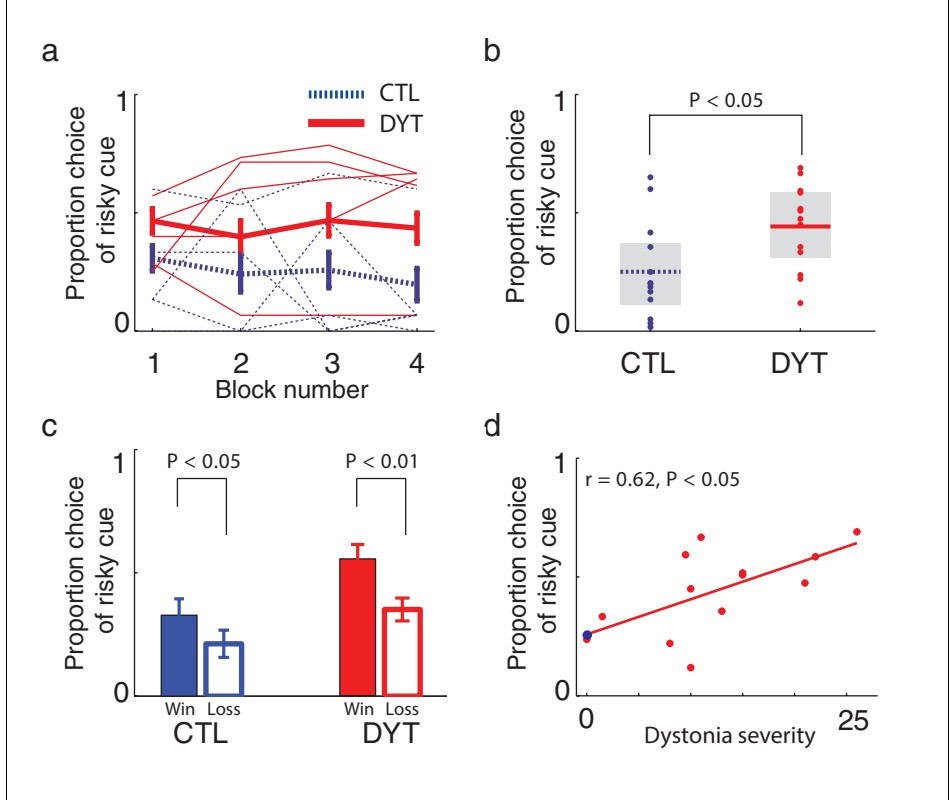

**Figure 3.** Risk taking in DYT1 dystonia patients as compared to healthy sex- and age-matched controls. (a) Mean proportion (± s.e.m) of choosing the risky 0/10¢ cue over the sure 5¢ cue (15 trials per block) in each of the groups. DYT1 dystonia patients (red solid) were less risk-averse than controls (blue dashed). Results from several randomly-selected participants are plotted in the background to illustrate within-participant fluctuations in risk preference over the course of the experiment, presumably driven by ongoing trial-and-error learning. (b) Overall percentage of choosing the risky 0/10¢ cue throughout the experiment. Horizontal lines denote group means; grey boxes contain the 25th to 75th percentiles. DYT1 dystonia patients showed significantly more risk-taking behavior than healthy controls. (c) Proportion of choices of the risky 0/10¢ cue over the sure 5¢ cue, divided according to the outcome of the previous instance in which the risky cue was selected. Both controls and DYT1 dystonia patients chose the risky 0/10¢ cue significantly more often after a 10¢ 'win' than after a 0¢ 'loss' outcome, demonstrating the effect of previous outcomes on the current value of the risky 0/10¢ cue due to ongoing reinforcement learning. Error bars: s.e.m. The effect of recent outcomes on the propensity to choose the risky option was evident throughout the task, especially in the DYT group, and was seen after both free choice and forced trials (*Figure 3—figure supplement 1*), suggesting that participants continuously updated the value of the risky cue based on feedback, and used this learned value to determine their choices. (d) Risk taking was correlated with clinical severity of dystonia (Fahn-Marsden dystonia rating scale). The mean of the control group is denoted in blue for illustration purposes only. Interestingly, the regression line for DYT1 dystonia patients' risk preference intersected the ordinate (0 severity of symptoms) close to the mean risk preference of healthy controls.

The following figure supplements are available for figure 3:

**Figure supplement 1.** Learning about the risky cue continued throughout the task.

**Figure supplement 2.** Effects of ongoing learning in the simulated data.

**Figure supplement 3.** Sex of participants did not affect risk sensitivity in our task.

**Figure supplement 4.** Medication did not affect risk-sensitivity.

severe dystonia were more risk taking in our task (*Figure 3d*, Pearson's r = 0.62, df = 11, P < 0.05). Risky behavior was not significantly affected by sex (*Figure 3—figure supplement 3*) or the patient's regime of regular medication (*Figure 3—figure supplement 4*), and the relationship between risk taking and symptom severity held even when controlling for these factors (p < 0.05 for symptom severity when regressing risk taking on symptom severity, age and either of the two medications; including both medications in the model lost significance for symptom severity, likely due to the large number of degrees of freedom for such a small sample size; age and medication did not achieve significance in any of the regressions).

To test whether increased risk-taking in DYT1 dystonia could be explained by asymmetry in the effects of positive and negative prediction errors on corticostriatal plasticity, we modeled participants' choice data using an asymmetric reinforcement-learning model (see methods) where the learning rate ($\eta$) is modulated by $(1 + \kappa)$ when learning from positive prediction errors and by $(1 - \kappa)$ when the prediction error is negative (also called a 'risk-sensitive' reinforcement learning model; *Mihatsch and Neuneier, 2002*; *Niv et al., 2012*). Our model also included an inverse-temperature parameter ($\beta$) controlling the randomness of choices. This approach exploits fluctuations in each individual's propensity for risk taking (see *Figure 3a*) as they update their policy based on the outcomes they experience, to recover the learning rate $\eta$ and learning asymmetry $\kappa$ that provide the best fit to each participant's observed behavior.

First, we tested whether the asymmetric-learning model is justified, that is, whether it explains participants' data significantly better than the classical reinforcement-learning model with only learning-rate and inverse-temperature parameters. The results showed that the more complex model was

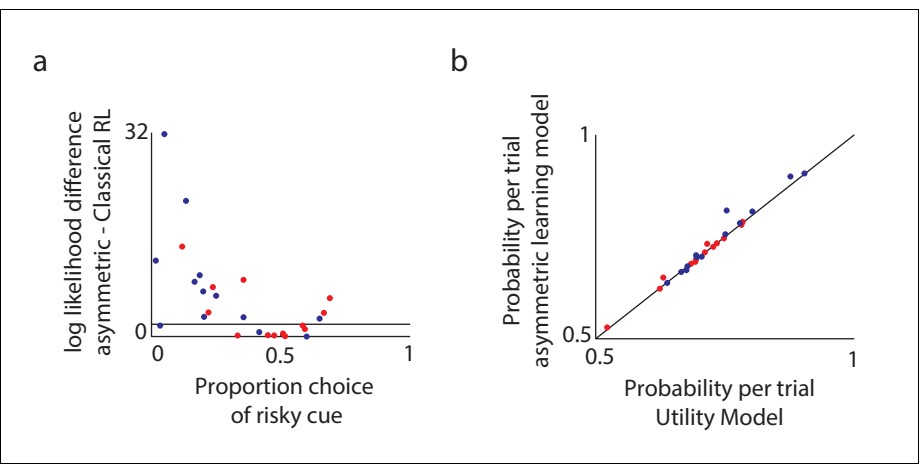

**Figure 4.** Model comparison supports the asymmetric learning model. We compared three alternative models in terms of how well they fit the experimental data. (**a**) To compare the asymmetric learning model with the classical (symmetric) reinforcement learning (RL) model, we used the likelihood ratio test, which is valid for nested models. Plotted are the log likelihood differences between the asymmetric learning model and the classical RL model. Black line: the minimal difference above which there is a P < 0.05 chance that the additional parameter improves the behavioral fit, as tested via a likelihood ratio test for nested models (dots above this line support the asymmetric learning model). For the majority of participants (16 out of 26; DYT 6, CTL 10) the more complex asymmetric model was justified (chi square test with df = 1, P < 0.05). In particular, and as expected based on *Niv et al. (2012)*, the asymmetric learning model was justified for participants who were either risk-averse or risk-taking, but not those who were risk-neutral. (**b**) The asymmetric learning model and the nonlinear utility model have the same number of free parameters and therefore could be compared directly using the likelihood of the data under each model. Plotted is the average probability per trial for the asymmetric learning model as compared to the nonlinear utility model, for healthy controls (blue dots) and patients with dystonia (red). Dots above the black line support the asymmetric learning model. The asymmetric learning model fit the majority of participants better than the utility model (15 out of 26; DYT 6, CTL 9) with large differences in likelihoods always in favor of the asymmetric model, and over the entire population the asymmetric learning model performed significantly better (paired one-tailed t-test on the difference in model likelihoods, t = 1.92, df = 25, P < 0.05).

justified for the majority of participants (16 out of 26 participants; DYT 6, CTL 19), and in particular, for participants who were risk seeking or risk taking (but not risk-neutral; *Figure 4a*).

We then compared the individually fit parameters of the asymmetric model across the two groups. We found significant differences between the groups in the learning asymmetry parameter (DYT $-0.05 \pm 0.27$, CTL $-0.34 \pm 0.27$, Mann-Whitney $z = -2.51$, df = 24, P < 0.05), but no differences in the other two parameters (learning rate DYT1 $0.25 \pm 0.19$, CTL $0.14 \pm 0.11$, Mann-Whitney $z = 1.33$, df = 24, P = 0.18; inverse temperature DYT $0.68 \pm 0.37$, CTL $0.93 \pm 0.47$, Mann-Whitney $z = -1.18$, df = 24, P = 0.23). Thus patients' behavior was consistent with enhanced learning from positive prediction errors and reduced learning from negative prediction errors as compared to healthy controls, despite the overall rate of learning and the degree of noise in choices (modeled by the inverse temperature parameter) being similar across groups. A significant correlation was also observed between the learning asymmetry parameter and the severity of dystonia (Pearson's $r = 0.64$, df = 11, P < 0.05).

One alternative explanation for our results is that the nonlinearity of subjective utility functions (*Kahneman and Tversky, 1979*) for small amounts of money is different between DYT1 dystonia patients and controls. However, replicating previous results from a healthy cohort (*Niv et al., 2012*), formal model comparison suggested that choice behavior in our task is significantly better explained by the asymmetric-learning model above (*Figure 4b*). Moreover, the impetus for our experiment was an *a priori* hypothesis regarding risk sensitivity as a consequence of asymmetric learning, based on findings from the mouse model of DYT1 dystonia, which has no straightforward equivalent interpretation in terms of nonlinear utilities. We note also that strongly nonlinear utilities in the domain of small payoffs such as those we used here are generally unlikely (*Rabin and Thaler, 2001*), again suggesting that risk sensitivity is more likely to arise in our experiment from asymmetric learning. Another alternative explanation for behavior in our task, is a win-stay lose-shift strategy that is perhaps utilized to different extent by DYT1 patients and controls. However, this model, equivalent to the classical reinforcement-learning model with a learning rate of 1 and only an inverse temperature parameter, fit 25 out of 26 participants' data considerably worse than the asymmetric learning model, and therefore was not investigated further.

## Discussion

We demonstrated that DYT1 dystonia patients and healthy controls have different profiles of risk sensitivity in a trial-and-error learning task. Our results support the dominant model of reinforcement learning in the basal ganglia, according to which prediction-error modulated LTP and LTD in corticostriatal synapses are responsible for changing the propensity to repeat actions that previously led to positive or negative prediction errors, respectively. Similar to Parkinson's disease, at first considered a motor disorder but now recognized to also cause cognitive and learning abnormalities, it appears that DYT1 dystonia is not limited to motor symptoms (*Fiorio et al., 2007*; *Heiman et al., 2004*; *Molloy et al., 2003*; *Stamelou et al., 2012*), and specifically, that the suspected altered balance between LTP and LTD in this disorder has overt, readily measurable effects on behavior.

DYT1 dystonia and Parkinson's disease can be viewed as complementary models for understanding the mechanisms of reinforcement learning in the human brain. In unmedicated Parkinson's disease patients, learning from positive prediction errors is impaired due to reduced levels of striatal dopamine that presumably signal the prediction errors themselves, whereas learning from negative prediction errors is intact (*Frank et al., 2004*; *Rutledge et al., 2009*). This impairment, and the resulting asymmetry that favors learning from negative prediction errors, can be alleviated using dopaminergic medication (*Frank et al., 2004*; *Shohamy et al., 2004*). DYT1 dystonia patients, on the other hand, seem to have intact striatal dopamine signaling (*Balcioglu et al., 2007*; *Dang et al., 2006*; *Grundmann et al., 2007*; *Zhao et al., 2008*), but altered corticostriatal LTP/LTD that favors learning from positive prediction errors.

Our *a priori* predictions were based on a simplified model of the role of corticostriatal LTP and LTD in reinforcement learning, and the entire picture is undoubtedly more complex. Controversies regarding the functional relationship between the direct and indirect pathways of the basal ganglia (*Calabresi et al., 2014*; *Cui et al., 2013*; *Kravitz et al., 2012*) and the large number of players taking part in shaping synaptic plasticity (*Calabresi et al., 2014*; *Shen et al., 2008*) make it hard to pin down the precise mechanism behind reinforcement learning. Indeed, the DYT1 mouse model has

also been linked to impaired plasticity in the indirect pathway due to D2 receptor dysfunction (*Beeler et al., 2012*; *Napolitano et al., 2010*; *Wiecki et al., 2009*), which can lead to abnormal reinforcement (*Kravitz et al., 2012*).

In any case, our finding are compatible with the prominent 'Go'/'NoGo' model of learning and action selection in the basal ganglia (*Frank et al., 2004*) that incorporates opposing directions of plasticity in the direct and indirect pathways (*Collins and Frank, 2014*). In particular, current evidence suggests that corticostriatal LTP following positive prediction errors and LTD following negative prediction errors occur in D1 striatal neurons (direct pathway), whereas plasticity in D2-expressing neurons (indirect pathway) is in the opposite direction (*Kravitz et al., 2012*; *Shen et al., 2008*). As the direct pathway supports choice ('Go') while the indirect pathway supports avoidance ('NoGo'), under this implementation of reinforcement learning both types of learning eventually lead to the same behavioral outcome: a positive prediction error increases the probability that the action/choice that led to the prediction error would be repeated in the future, and vice versa for negative prediction errors. As such, at the algorithmic level in which our asymmetric learning model was cast, the differences we have shown between dystonia patients and controls would still be expected to manifest behaviorally through diminished risk-aversion in dystonia patients.

In particular, our results are compatible with several alternative abnormalities in corticostriatal plasticity in DYT1 dystonia: (a) Abnormally strong LTP/weak LTD in D1-expressing striatal neurons only, with plasticity in the indirect pathway being intact; in this case, learning in the direct pathway would exhibit the abnormal asymmetries we argue for, whereas the indirect pathway would learn as normal. (b) Abnormally strong LTP/weak LTD in D1-expressing striatal neurons and the opposite pattern, abnormally strong LTD and/or weak LTP in D2-expressing striatal neurons of the indirect pathway in DYT1 dystonia. As a result, a positive prediction error would generate extra strong positive learning in the Go pathway, and a similarly large decrease in the propensity to avoid this stimulus due to activity in the 'NoGo' pathway. Conversely, learning from negative prediction errors would generate relatively little decrease in the propensity to 'Go' to the stimulus and little increase in the propensity to 'NoGo'. In both cases, the effect on both pathways would be in the same direction as is seen in the behavioral asymmetry. (c) Finally, abnormalities may exist in both pathways in the same direction (stronger LTP and weaker LTD), but with a larger effect on LTP as compared to LTD. In this case, a positive prediction error would increase 'Go' activity considerably, but not decrease 'NoGo' activity to the same extent. Negative prediction errors, on the other hand, would increase 'NoGo' propensities while decreasing 'Go' propensities to a lesser extent. This type of asymmetry can explain why the rodent studies suggested almost absent (not only weaker) LTD, but nevertheless, patients did not behave as if they did not learn at all from negative prediction errors. Unfortunately, our model and behavioral results cannot differentiate between these three options. We hope that future data, especially from transgenic DYT1 rodents, will clarify this issue.

Relative weighting of positive and negative outcomes shapes risk-sensitivity in tasks that involve learning from experience. Humans with preserved function of the basal ganglia have been shown to be risk-averse in such tasks. We showed that patients with DYT1 dystonia are more risk-neutral, a rational pattern of behavior given our reward statistics, and in such tasks in general. While this type of behavior may offer advantages under certain conditions, it may also contribute to impaired reinforcement learning of motor repertoire and fixation on actions that were once rewarded. In any case, these reinforcement-learning manifestations of what has been considered predominantly a motor disease provide support for linking corticostriatal synaptic plasticity and overt trial-and-error learning behavior in humans.

## Materials and methods

### Subjects

Fourteen participants with genetically-proven (c.907_909delGAG) (*Ozelius et al., 1997*) symptomatic DYT1 dystonia were recruited through the clinics for movement disorders in Columbia University and Beth Israel Medical Centers in New York and through publication in the website of the Dystonia Medical Research Foundation. Exclusion criteria included age younger than 18 or older than 50 years old and deep brain stimulation or other prior brain surgeries for dystonia. A single patient was excluded from further analysis due to choosing the left cue in 100% of trials. Thirteen age- and sex-

matched healthy participants were recruited among acquaintances of the DYT1 patients and from the Princeton University community. Healthy control participants were not blood relatives of patients with dystonia and did not have clinical dystonia. All patients and healthy controls had at least 13 years of education.

Nine DYT1 dystonia patients took baclofen (n = 6, daily dose 66.7 ± 28.0 mg, range 30–100 mg) and/or trihexyphenidyl (n = 7, daily dose 30.9 ± 25.8 mg, range 12–80 mg) for their motor symptoms. In order to reduce possible effects of medication, patients were tested before taking their scheduled dose. The median time interval between the last dose of medication and testing was 7.5 hr for baclofen (range 1–20 hr) and 13 hr for trihexyphenidyl (range 1–15 hr). Given that the reported plasma half-life times of baclofen is 6.8 hr (*Wuis et al., 1989*) and of trihexyphenidyl is 3.7 hr (*Burke and Fahn, 1985*), three patients were tested within the plasma half life of the last dose of their medication. Finally, we could not find correlation between sex of participants (*Figure 3—figure supplement 1*) or medication doses (*Figure 3—figure supplement 4*) and relevant behavioral outcomes.

## Procedure

All participants gave informed consent and the study was approved by the Institutional Review Boards of Columbia University, Beth Israel Medical Center, and Princeton University. Clinical scale of dystonia severity was scored immediately after consenting by a movement-disorders specialist (DA), using the Fahn-Marsden dystonia rating scale (*Burke et al., 1985*). This scale integrates the number of involved body parts, the range of actions that induce dystonia and the severity of observed dystonia. One patient was scored 0 since dystonia was not clinically observed on the day of her testing.

Prior to, and after completing the reported task, all participants performed a short (8–9 min) unrelated auditory discrimination task (*Baron, 1973*) (results not reported here) that was not associated with any monetary reward. Participants were informed that the two tasks were not related.

## Behavioral task

Four different pseudo-letters served as cues ('slot machines') and were randomly allocated to four payoff schedules: sure 0¢, sure 5¢, sure 10¢, and one variable-payoff 'risky' stimulus associated with equal probabilities of 0¢ or 10¢ payoffs. Participants were not informed about the payoffs associated with the different cues and had to learn them from trial and error.

Two types of trials were pseudo-randomly intermixed: In 'choice trials', two cues were displayed (left and right location randomized), and the participant was instructed to select one of the two cues by pressing either the left or right buttons on a keyboard. The cue that was not selected then disappeared and the payoff associated with the chosen cue was displayed for 1 s. After a variable (uniformly distributed) inter-trial interval of 1–2 s, the next trial began. In 'forced trials', only one cue was displayed on either the left or right side of the screen, and the participant had to indicate its location using the keyboard to obtain its associated outcome. All button presses were timed out after 1.5 s, at which time the trial was aborted with a message indicating that the response was 'too slow' and the inter-trial interval commenced. Participants were instructed to try to maximize their winnings and were paid according to their actual payoffs in the task. On-screen instructions for the task also informed participants that payoffs depended only on the 'slot machine' chosen, not on its location or on their history of choices. Throughout the experiment, to minimize motor precision requirements, any of keys E, W, A, Z, X, D, and S (on the left side of the keyboard) served as allowable response buttons for choosing the left cue and any of keys I, O, L, <, M, J and K (on the right side of the keyboard) served as allowable response buttons for choosing the right cue. Each set of response keys was marked with stickers of different colors (blue for left keys and red for right keys) to aid in their visual identification.

Participants were first familiarized with the task and provided with several observations of the cue–reward mapping in a training phase that included two subparts. The first part involved 16 pseudo-randomly ordered forced trials (four per cue). The second part comprised 10 pseudo-randomly ordered choice trials (two of each of five types of choice trials: 0¢ versus 5¢, 5¢ versus 10¢, 0¢ versus 0/10¢, 5¢ versus 0/10¢ and 10¢ versus 0/10¢).

Before the experimental task began, on-screen instructions informed subjects that they would encounter the same cues as in the training phase. They were briefly reminded of the rules and

encouraged to choose those 'slot machines' that yielded the highest payoffs, as they would be paid their earnings in this part. The task then consisted of 300 trials (two blocks of 150 trials each, with short breaks after every 100 trials of the experiment), with choice and forced trials randomly intermixed. Each block comprised of 30 'risk' choice trials involving a choice between the 5¢ cue and the 0/10¢ cue, 20 choice trials involving each of the pairs 0¢ versus 0/10¢ and 10¢ versus 0/10¢, 15 choice trials involving each of the pairs 0¢ versus 5¢ and 5¢ versus 10¢, 14 forced trials involving the 0/10¢ cue and 12 forced trials involving each of the 0¢, 5¢ and 10¢ cues. Trial order was pseudo-randomized in advance and was similar between patients and between blocks. Payoffs for the 0/10¢ cue were counterbalanced such that groups of eight consecutive choices of the risky cue included exactly four 0¢ payoffs and four 10¢ payoffs. All task events were controlled using MATLAB (MathWorks, Natick, MA) PsychToolbox (*Brainard, 1997*).

Our modeling and quantification of the effects of abnormal learning from prediction errors rest solely on the risky cue, for which learning presumably continued throughout the experiment. However, one potential worry is that participants did not use trial-and-error learning to evaluate this cue, but rather 'guessed' its value using a cognitive system (as in rule-based learning). To evaluate this possibility, we tested for a difference in the propensity to choose the risky cue after a previous win or a loss, throughout the task (see *Figure 3c* and *Figure 3—figure supplement 1*).

## Modeling

To test the hypothesis that increased risk-taking in DYT1 dystonia was due to an enhanced effect of positive prediction errors and a weak effect of negative prediction errors, we modeled participants' choice data using an asymmetric reinforcement learning model (also called a risk-sensitive temporal difference reinforcement learning (RSTD) model) (*Mihatsch and Neuneier, 2002*; *Niv et al., 2012*). The learning rule in this model is

$$V^{new}(cue) = V^{old}(cue) + \eta \cdot \delta \cdot (1 \pm \kappa)$$

where *V(cue)* is the value of the chosen cue, $\delta = r - V^{old}(cue)$ is the prediction error, that is, the difference between the obtained reward r and the predicted reward $V^{old}$(cue), $\eta$ is a learning-rate parameter and $\kappa$ is an asymmetry parameter that is applied as $(1 + \kappa)$ if the prediction error is positive ($\delta > 0$) and as $(1 - \kappa)$ if the prediction error is negative ($\delta < 0$). This model is fully equivalent to a model with two learning rate parameters, one for learning when prediction errors are positive and another for learning when prediction errors are negative. Following common practice, we also assumed a softmax (or sigmoid) action selection function:

$$p(A) = \frac{e^{\beta V(A)}}{e^{\beta V(A)} + e^{\beta V(B)}}$$

where *p(A)* is the probability of choosing cue A over cue B, and $\beta$ is an inverse-temperature parameter controlling the randomness of choices (*Niv et al., 2012*).

We fit the free parameters of the model ($\eta$, $\kappa$, and $\beta$) to behavioral data of individual participants, using data from both training and test trials (total of 326 trials) as participants learned to associate cues with their outcomes from the first training trial. Cue values were initialized to 0. We optimized model parameters by minimizing the negative log likelihood of the data given different settings of the model parameters using the Matlab function "fminunc". The explored ranges of model parameters was [0,1] for the learning-rate parameter, [−10,10] for the learning-asymmetry parameters, and [0–30] for the inverse-temperature parameter. To avoid local minima, for each participant we repeated the optimization 5 times from randomly chosen starting points, keeping the best (maximum likelihood) result. This method is commonly used for temporal difference learning models and is known to be well-behaved (*Niv et al., 2012*).

Previous work has shown that the asymmetric learning model best explains participants' behavior in our task (*Niv et al., 2012*). To replicate those results in our sample population, we compared the asymmetric learning model to three other alternative models. The first was a classical reinforcement learning model with no learning asymmetry $V^{new}(cue) = V^{old}(cue) + \eta[R - V^{old}(cue)]$. The second alternative model was based on the classical nonlinear (diminishing) subjective utility of monetary rewards. The idea is that the 10¢ reward may not be subjectively equal to twice the 5¢ reward, therefore engendering risk-sensitive choices in our task. We thus defined learning in a

nonlinear utility model as $V^{new}(cue) = V^{old}(cue) + \eta\left[U(R) - V^{old}(cue)\right]$, where $U(R)$ is the subjective utility of reward $R$. Without loss of generality, we parameterized the utility function over the three possible outcomes (0¢, 5¢ or 10¢) by setting $U(0) = 0$, $U(5) = 5$ and $U(10) = a \times 10$, where the parameter $a$ could be larger, equal to or smaller than 1, and was fit to the data of each participant separately. If the effect of a loss is larger than that of a gain of the same magnitude, $a$ should be smaller than 1. Finally, we tested a win-stay-lose-shift strategy model that is equivalent to the classic reinforcement learning model with a learning rate of 1. All models used the softmax choice function with an inverse temperature parameter $\beta$. The parameters of each of the models were fit to each participant's data as was done for the asymmetric learning model.

## Statistical analysis

Because the relevant sets of data were not normally distributed (tested using a Kolmogorov-Smirnov test, $P < 0.05$), we analyzed the data using the nonparametric Mann-Whitney U test to compare two populations, Wilcoxon signed-rank test for repeated measures tests, and Friedman's test for nonparametric one-way repeated measures analysis of variance by ranks. All statistical tests were two-sided unless otherwise specified.

## Acknowledgements

This research was supported in part by the Parkinson's Disease Foundation (DA), the NIH Office of Rare Diseases Research through the Dystonia Coalition (DA) and the National Institute for Psychobiology in Israel (DA), a Sloan Research Fellowship to YN, NIH grant R01MH098861 (AR and YN) and Army Research Office grant W911NF-14-1-0101 (YN & AR). We are grateful to Hagai Bergman, Reka Daniel, Nathaniel Daw, Stanley Fahn, Ann Graybiel, Elliot Ludvig, Rony Paz, Daphna Shohamy, Nicholas Turk-Browne and Jeff Wickens for very helpful comments on previous versions of the manuscript.

## Additional information

### Funding

| Funder | Grant reference number | Author |
| --- | --- | --- |
| Parkinson's Disease Foundation | | David Arkadir |
| National Institutes of Health | NIH Office of Rare Diseases Research through the Dystonia Coalition | David Arkadir |
| National Institute for Psychobiology in Israel, Hebrew University of Jerusalem | | David Arkadir |
| Alfred P. Sloan Foundation | Sloan Research Fellowship | Yael Niv |
| National Institute of Mental Health | R01MH098861 | Angela Radulescu Yael Niv |
| Army Research Office | W911NF-14-1-0101 | Angela Radulescu Yael Niv |

The funders had no role in study design, data collection and interpretation, or the decision to submit the work for publication.

### Author contributions

DA, AR, Conception and design, Acquisition of data, Analysis and interpretation of data, Drafting or revising the article; DR, NL, SBB, Facilitated access to patient population, Assistance with patient recruitment, Drafting or revising the article; PM, Analysis and interpretation of data, Drafting or revising the article; YN, Conception and design, Analysis and interpretation of data, Drafting or revising the article

## Author ORCIDs

Yael Niv, http://orcid.org/0000-0002-0259-8371

## Ethics

Human subjects: All participants gave informed consent and the study was approved by the Institutional Review Boards of Columbia University, Beth Israel Medical Center, and Princeton University.

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
