## [Decision Letter]

Thank you for submitting your article "DYT1 dystonia increases risk taking in humans" for consideration by *eLife*. Your article has been reviewed by three peer reviewers, and the evaluation has been overseen by a Reviewing Editor and Timothy Behrens as the Senior Editor.

The reviewers have discussed the reviews with one another and the Reviewing Editor has drafted this decision to help you prepare a revised submission. The reviewers found the behavioral findings interesting. However, they indicate that the model presented needs to be better validated, and that alternative models should be considered to explain the data. For example, given the current claims it is important to verify that the risk-taking effect is demonstrably a learning effect rather than a choice-like effect or a WSLS type effect that would not implicate striatal plasticity. To be clear, this was the key concern raised in the reviewer discussion and the reviewers were clear that publication of the study would be contingent on the outcome of these analyses.

Furthermore, in the current version of the study, there is no evidence that the changes in corticostriatal plasticity mentioned are responsible for the risk taking behavior. So that interpretation should be toned down, unless new evidence is presented. Below are the detailed comments, which should help in preparing a revised version.

Reviewer #1:

I enjoyed reading this relatively brief and focused test of the role of dopamine in learning in human subjects. Several previous reports have exploited the fact that the impact of diminished dopamine on learning can be investigated in Parkinson's disease patients (Frank et al., Science, 2004, Shohamy et al., Brain, 2004; Rutledge et al., J. Neurosci., 2009) but the novelty of this report is that it examines the impact of a dopaminergic/striatal change occurring in a type of dystonia, DYT1 dystonia, that is, in a sense, the opposite way round (in the sense that there is an increase in LTP/decrease in LTD in transgenic animals). Perhaps the only other way to conduct a test of this sort is by examining the impact of dopamine increases by L-DOPA treatment (e.g. Pessiglione et al., Nature, 2006).

The task used is one that has been previously used by one of the authors (Niv et al., J. Neuroscience, 2012). The previous report also justifies the modeling approach taken. I therefore had very few questions. I wondered, however, about the following minor points.

1) In the last paragraph of the Introduction. Although I agree that the results are interesting I was not always quite sure precisely how the authors intended to frame them. Perhaps changing or adding a few words in the Introduction or Discussion might help provide the final clarification needed. The current version emphasizes that the dystonic condition is associated with a change in LTP/LTD balance and emphasizes that this might lead to a change in learning from positive prediction errors. By contrast when discussing studies in Parkinson's patients they seem to emphasize dopaminergic changes. Is it not the case that changes in dopamine (or its balance with other neurotransmitters) occur both in Parkinson's patients and dystonic patients and that as a consequence both will be expected to lead to changes in LTP /LTD ratios. Or is there no evidence of a change in LTP/LDT in Parkinson's disease?

2) While I understand that most patients with this particular condition have normal general intelligence I wondered if there was any information about intelligence/education matching in the control and patient groups reported here?

3) In the fourth paragraph of the Discussion. It is stated that "Current methodologies cannot yet bridge the gap between cellular level processes (LTP/LTD) and behavior at the level of an animal, let alone a human." Arguably, however, there are techniques that can be used to induce increments and decrements in the strengths of specific connexions even in awake, behaving human subjects (Buch et al., 2011, J. Neuroscience, Johnen et al., *eLife*, 2015) although their use is limited to the cortex.

Reviewer #2:

This study investigates risk-taking in a reinforcement learning experiment in 13 patients with dystonia and matched controls. Subjects learned to select between pairs of cues (or are forced to pick one). Three cues lead deterministically to 0, 5 or 10c, and the last one to 0 or 10c with equal probability. Risk was assessed by the choice between the 5c and the probabilistic cues, which have same expected value. The authors show that controls are significantly more risk-averse in this task than patients, who appear to be risk neutral. They show evidence that this is not due to patients choosing randomly, but that they are sensitive to previous reward and thus presumably learning the value over time. They argue that the behavioral difference can be accounted for by assuming variable learning rates for positive vs. negative prediction errors, with a greater asymmetry for patients than controls. The authors hypothesize, based on a rodent model showing abnormally high LTP and low LTD, that the observed behavioral effects thus reflect differences in synaptic plasticity.

Overall, this paper is well written and interesting. There are some limitations to the study.

1) The authors overstate the implications of this study in the Abstract, main text and Discussion. For example, the Discussion states that this "may provide the first link between synaptic plasticity and overt learning behavior in humans". This is a stretch: for one, based on the literature they present, there is no direct evidence that dyt1 patients' impairment is indeed a plasticity impairment. Furthermore, the behavioral evidence tying risk taking to learning here is also weak (see next point). These results may provide more evidence, of a different nature, but of a similar degree of "directness" as existing evidence, for example linking dopamine dependent plasticity individual differences to learning behavior in humans.

2) Model validation. Despite the authors' careful effort to validate it, the RSTD is still not very convincing in this data set.

A) While it fits better over the whole group, it does not seem to be the case within the patient group (6/13). This is a problem, as it may indicate that the fit parameters don't reflect important aspects of the behavior within this group, rendering interpretation difficult.

B) This may be a limitation of the task: Dyt1 patients appear to be in average risk neutral, such that a task in which the only risk comparison relies on equal average value will render the RSTD not identifiable beyond TD.

C) An important validation of the model would be to confirm with simulations that it can reproduce behavioral pattern 3c. This is an important, model-independent data point supporting the learning interpretation, and the model should at minima be able to capture this qualitative phenomenon.

D) As the authors point out, there is nearly no learning needed in this task. A natural model to consider would thus be a non-learning utility model, parameterized with the same a parameter as the learning utility model presented here, but assuming known probabilities. While 3c hints at a potential learning effect, it could also reflect a non-learning win-stay lose-shift strategy. Investigating whether this combination accounts for behavior better or worse than RSTD would be an important point in ensuring whether the learning interpretation is valid.

3) If the RSTD model is validated, it would be useful to also present results relating to the learning rates directly, not just the difference index, as authors attempt an interpretation in terms of LTP/and or LTD.

4) It would also be important to know whether previous rewards have differential effects depending on whether the trial was a free or forced choice trial (e.g. Figure 3). Recent work (Cockburn et al) has shown that value learning, especially from positive prediction errors, differed between these conditions, and if patients and controls had baseline differences in risk taking for non-learning reasons, this would lead patients to experience more free-choice risky trials than controls, potentially biasing learning. This might be important to rule out.

5) It would be interesting to report patients' behavior in absolute, not only relative to controls. It appears that they are in average risk neutral, which is not a suboptimal or irrational thing to do in this task. It might be interesting to discuss why their impairment appears to "correct an imbalance" that is seen in healthy controls, contrary to what is usually observed in patient studies.

Reviewer #3:

Building on animal work showing that DYT1 dystonia animal models are associated with exaggerated LTP and diminished LTD, the authors outline an experiment testing for learning abnormalities in human DYT1 dystonia patients. The authors hypothesize that DYT1 patients will show a positive learning bias owing to increased LTP and/or reduced LTD. Consistent with their hypothesis, DYT1 patients do indeed show atypical responsivity to outcomes in that they do not exhibit risk aversion as did matched controls.

The authors correctly emphasize the chasm between synaptic modification and behavior. The application of genetically linked animal models and human disease states in concert with an algorithmic description of a learning mechanism offers an exciting and constructive path forward. However, this depends critically on a shared mechanism between the animal model and the human brain. I cannot speak to the quality of the animal models or the nature of DYT1 dystonia; however, the text leaves some question as to the shared commonality. The authors point to atypical LPT/LTD in animal models, but abnormal dopamine/acetylcholine transmission in humans. Given the manuscript's emphasis on lessening the gap between synapse and behavior, I feel that the manuscript could benefit from more support linking animal and human disorders (perhaps via behavioral patterns in the animal models, mechanisms through which medications operate etc.).

Of greater concern is whether these results truly inculpate a reinforcement learning mechanism. As outlined out by the authors, patients exhibited a strong propensity to pick the risky option again following a win but avoided it following a loss, which is argued to demonstrate outcome sensitivity. But, this response pattern does not necessitate a reinforcement learning strategy. These data also appear to be consistent with a win-stay/lose-shift strategy (WS/LS). Given the 50% chance of reward on the risky stimulus, a WS/LS strategy is consistent with the reported lack of risk aversion. Furthermore, the model generated response pattern illustrated in Figure 1 (bottom panel), suggests that a reinforcement learning strategy becomes increasingly risk averse. There does appear to be some trend of increasing risk aversion in controls, but not so for the patient group. Given that there are no clear response patterns that demonstrate a reinforcement learning strategy (i.e. there are no learning curves etc.), I feel that the results would be more interpretable if a WS/LS model was also included in the analysis and Discussion.

Could observed effects in the patient group be driven by medication withdrawal? The authors offer helpful discussion of medication half-life, but this only serves to demonstrate that effects are probably not driven by the medication itself.

What were the model parameter ranges explored while fitting the data? Within subject variance (Figure 3) seem to indicate considerably more variance in the control group (though I understand this to be a subset of the dataset), so I am a bit surprised by the apparent lack of effect in the inverse temperature parameter.

---

## [Author Response]

*Reviewer #1:*

1) In the last paragraph of the Introduction. Although I agree that the results are interesting I was not always quite sure precisely how the authors intended to frame them. Perhaps changing or adding a few words in the Introduction or Discussion might help provide the final clarification needed. The current version emphasizes that the dystonic condition is associated with a change in LTP/LTD balance and emphasizes that this might lead to a change in learning from positive prediction errors. By contrast when discussing studies in Parkinson's patients they seem to emphasize dopaminergic changes. Is it not the case that changes in dopamine (or its balance with other neurotransmitters) occur both in Parkinson's patients and dystonic patients and that as a consequence both will be expected to lead to changes in LTP /LTD ratios. Or is there no evidence of a change in LTP/LDT in Parkinson's disease?

We thank the reviewer for this comment and apologize for the confusion. Indeed, there is no evidence of altered dopaminergic signaling in dystonia whereas plasticity itself seems intact in Parkinson’s disease. Therefore, the two disorders model alterations in different parts of the hypothesized mechanism of reinforcement learning.

Dopamine release in the striatum is preserved in both patients with DYT1 dystonia and rodent models of the disease. Autopsies in human with the disease did not reveal any evidence for loss of midbrain dopaminergic neurons (Furukawa et al., 2000; Rostasy et al., 2003) and striatal dopamine levels are normal (Augood et al., 2002; Furukawa et al., 2000). Also, in Rodent models these levels are normal (Balcioglu et al., 2007; Dang et al., 2006; Grundmann et al., 2007; Zhao et al., 2008). Lack of response of patients with DYT1 dystonia to either dopaminergic agonists or antagonists furthermore supports the assumption that striatal levels of dopamine do not play a role in DYT1 dystonia. The physiological importance of certain impairments such as reduced dopamine release triggered by amphetamine (Balcioglu et al., 2007) and increase dopamine turnover (Zhao et al., 2008) is still unknown.

This stands in contrast with Parkinson's disease in which dopamine deficiency (due to progressive death of dopamine neurons in the substantia nigra pars compacta) is the uncontested main player. Indeed, there is no evidence for impaired corticostriatal plasticity (LTP/LTD) in Parkinson’s disease, as the therapeutic effect of increasing dopamine levels in the striatum in this disorder can partly attest to (Thiele et al., 2014).

To summarize, our claim is that while impairments in either striatal dopamine and corticostriatal plasticity can, according to models of reinforcement learning in the basal ganglia, result in similar behavioral outcomes, one finding does not fall out from the other, and they each need to be tested separately. In this sense, our findings are novel and provide non-redundant support for the reinforcement learning model. Based on the Reviewer‘s suggestion we have now modified the Introduction of our revised manuscript so as to more clearly state how our study, that investigates the effects of presumed altered plasticity (but intact dopamine signaling) is different from (but related to) findings from Parkinson’s disease where plasticity is intact but dopamine signaling is altered:

“Dopamine’s role as a reinforcing signal for trial-and-error learning is supported by numerous findings (Pessiglione et al., 2006; Schultz et al., 1997; Steinberg et al., 2013), including in humans, where Parkinson’s disease serves as a human model for altered dopaminergic transmission (Frank et al., 2004). […] In particular, our predictions stem from considering the effects of intact prediction errors on an altered plasticity mechanism that amplifies the effect of positive prediction errors (i.e., responds to positive prediction errors with more LTP than would otherwise occur in controls) and mutes the effects of negative prediction errors (that is, responds with weakened LTD as compared to controls).”

We have also modified the Discussion to make this point clearer:

“DYT1 dystonia and Parkinson's disease can be viewed as complementary models for understanding the mechanisms of reinforcement learning in the human brain.. […] DYT1 dystonia patients, on the other hand, seem to have intact striatal dopamine signaling, but altered corticostriatal LTP/LTD that favors learning from positive prediction errors.”

2) While I understand that most patients with this particular condition have normal general intelligence I wondered if there was any information about intelligence/education matching in the control and patient groups reported here?

We agree with the Reviewer that this information is important. We matched the level of education between groups and all our patient had normal intelligence and at least 13 years of formal education. This information is given under Material and methods. We did not perform formal intelligence tests but confirmed that all subject fully understood the task.

3) In the fourth paragraph of the Discussion. It is stated that "Current methodologies cannot yet bridge the gap between cellular level processes (LTP/LTD) and behavior at the level of an animal, let alone a human." Arguably, however, there are techniques that can be used to induce increments and decrements in the strengths of specific connexions even in awake, behaving human subjects (Buch et al., 2011, J. Neuroscience, Johnen et al., eLife, 2015) although their use is limited to the cortex.

The Reviewer is absolutely correct that the introduction of different TMS protocols (such as paired-pulse stimuli) narrows the gap between LTP/LTD as demonstrated on the cellular levels and behaving organisms. We also agree with the Reviewer that this technique probably modulates cortical plasticity. We therefore deleted this sentence.

Reviewer #2:

1) The authors overstate the implications of this study in the Abstract, main text and Discussion. For example, the Discussion states that this "may provide the first link between synaptic plasticity and overt learning behavior in humans". This is a stretch: for one, based on the literature they present, there is no direct evidence that dyt1 patients' impairment is indeed a plasticity impairment. Furthermore, the behavioral evidence tying risk taking to learning here is also weak (see next point). These results may provide more evidence, of a different nature, but of a similar degree of "directness" as existing evidence, for example linking dopamine dependent plasticity individual differences to learning behavior in humans.

We thank the Reviewer for his comment, and agree that we might have overstated our findings. We have now adopted the Reviewer’s suggestion and attenuated our claim by modifying the relevant sentences in the manuscript. For example, in the Abstract, "implicating striatal plasticity" has been modified to "supporting striatal plasticity" and in the final sentence of the Discussion, "[…] may provide the first link between synaptic plasticity and overt learning behavior" has been modified to".[…] support the link between synaptic plasticity and overt learning behavior".

*2) Model validation. Despite the authors' careful effort to validate it, the RSTD is still not very convincing in this data set.*

*A) While it fits better over the whole group, it does not seem to be the case within the patient group (6/13). This is a problem, as it may indicate that the fit parameters don't reflect important aspects of the behavior within this group, rendering interpretation difficult.*

*B) This may be a limitation of the task: Dyt1 patients appear to be in average risk neutral, such that a task in which the only risk comparison relies on equal average value will render the RSTD not identifiable beyond TD.*

We thank the reviewer for this comment. We fully agree with the reviewer's claim that splitting the single learning-rate parameter of the classical TD model into two separate learning rate parameters (positive and negative) in our risk-sensitive temporal difference (RSTD) model is mainly justified for participant who are either risk-averse or risk-taking. The analysis presented in Figure 3—figure supplement 4 supports this claim. This analysis shows that the behavior of risk-averse or risk-taking participants in both groups was better described by our model.

Our behavioral task was designed to test our hypothesis that DYT1 patients will be less risk averse than controls due to relative overweighting of outcomes associated with positive prediction values. This hypothesis was indeed supported by the behavioral data—this is our main result, and the modeling was only used to illustrate and further clarify the (suggested) provenance of the behavioral findings. That is, we did not design the task for differentiating between models, but rather only to test the risk sensitivity of participants. It is for this reason that in our task we compared a risky and a non-risky option with similar mean values – we thought this was the most direct way to isolate the effect of risk on behavioral decision making.

Given the model-comparison results, an alternative way to frame the results in terms of the models is to say that DYT1 dystonia patients show no difference between learning from positive and negative prediction errors, and thus are better fit by a TD model with a single learning rate, in contrast to healthy controls. We believe that this framing is not more illuminating, and in fact, would suffer from the same criticism as only 7/13 of the patients are better fit with this model. That is, the patient group, as a whole, is “indifferent” to the two models. However, we respectfully disagree with the Reviewer that this means that the model parameters are perhaps unreliable – in participants better fit by the TD model with one learning rate, the RSTD model also showed that the two learning rates were similar, thus those parameters are as reliable. In particular, one way to quantify the similarity between the learning rates is as |η+−η−|η++η−, that is, the absolute difference between the two learning rates scaled by their overall magnitude. This similarity metric was < 0.2 for all participants who were better fit by the TD model, in either group (see Figure 5, circles denote subjects better fit by the RSTD model, stars denote subjects better fit by the simpler TD model).

Author response image 1.Learning rate similarity by participant.**DOI:**
http://dx.doi.org/10.7554/eLife.14155.011

All this having been said, our modeling exercise was not intended as a separate result, but only to aid in the interpretation of the main behavioral result. Indeed, in a previous version of the paper (that we ended up not submitting) we did not include the modeling at all. If the Reviewer and editors feel that the modeling is superfluous or otherwise distracting from the main result, we are happy to remove it altogether.

*C) An important validation of the model would be to confirm with simulations that it can reproduce behavioral pattern 3c. This is an important, model-independent data point supporting the learning interpretation, and the model should at minima be able to capture this qualitative phenomenon.*

We thank the Reviewer for this comment. Following the Reviewer’s suggestion, we simulated the behavioral pattern in Figure 3, using for the RSTD model the learning rates that were fit to each individual based on her/his behavior. As can be seen in Figure 3—figure supplement 2, the simulation qualitatively captured the observed pattern of behavior.

D) As the authors point out, there is nearly no learning needed in this task. A natural model to consider would thus be a non-learning utility model, parameterized with the same a parameter as the learning utility model presented here, but assuming known probabilities. While 3c hints at a potential learning effect, it could also reflect a non-learning win-stay lose-shift strategy. Investigating whether this combination accounts for behavior better or worse than RSTD would be an important point in ensuring whether the learning interpretation is valid.

We thank the reviewer for this important comment. The reviewer is correct that the results presents in Figure 3 indicate that participants continuously updated (i.e., learned) the value of the risky cue and updated their behavioral policy accordingly. While the learning requirement for non-risky cues was minimal, it is not clear to us how participants could have known the true probabilities for the risky stimulus absent learning, hence we did not test a model that did not include learning at all.

Inspired by the Reviewer’s comment, we now tested a win-stay-lose-shift (WSLS) model with learning – this is none other than the TD model with learning rate of 1 (that is, a model that chooses the risky stimulus after every win and avoids it after every loss, with a softmax parameter that allows for only a tendency, rather than absolute responding according to the predefined WSLS strategy). A likelihood ratio test showed that this model is inferior to the RSTD model for 25 out of the 26 participants (DYT = 12, CTR = 13; p < 0.05, Chi square test with df = 2), as seen in Figure 6 (points above the diagonal favor the RSTD model; red = DYT, blue = CTR).

Author response image 2.Comparison of risk sensitive (learning asymmetry) model and win-stay-lose-shift model.**DOI:**
http://dx.doi.org/10.7554/eLife.14155.012

Another, more “model-free,” way to show that the choices of the risky cue depend on reinforcement learning and not on a heuristic such as WSLS, is to test whether choices were sensitive not only to the most recent outcome for this cue, but also preceding outcomes. Indeed, as shown in Figure 7, choices in both groups differed depending on the outcome for the risky cue in the last two times it was chosen, as would be expected from reinforcement learning. That is, the tendency to choose this cue was highest after two “wins”, lowest after two “losses” and was intermediate for one “win” and one “loss”, with a more recent “win” contributing to a higher propensity to choose the cue again. These results are exactly as would be expected from a reinforcement learning model as this model computes the value of a cue as a recency-weighted average of the outcomes obtained for the cue. We did not test three trials back and further because the short length of our experiment and the few trials involving the risky cue limit our power once trials are divided into 8 combinations of wins and losses.

Author response image 3.Effect of outcomes of the past two trials on choices of the risky cue.**DOI:**
http://dx.doi.org/10.7554/eLife.14155.013

One other piece of information that bears on this question is that after completing the task, we asked participants to verbally associate the different cues with outcomes and their probabilities. Only 9 of the 26 participants (35%) estimated the monetary value of the risky cue correctly (see table below). Among the 21 participants who associated the risky cue with the correct outcomes (0 or 10¢), the mean estimated outcome was not significantly different across groups (DYT: N=10, 5.38 ± 1.37¢; CTL: N=11, 4.44 ± 0.98¢, p=0.09, t test). Moreover, there was no correlation between the reported value and behavioral risk taking (N=21, Pearson's r=0.26, p=0.26), suggesting that the explicit estimation of monetary value at the end of the task and choice behavior throughout the task were not tightly related.

**Reported outcome for risky cue****DYT1****CTL**Correct answer P(10) = p(0)45Answered that P(10)>P(0)41Answered that P(10)<P(0)25Could not estimate or associated the risky cue with incorrect 5 cent outcome (0/5/10 or 5/10)32Total1313

Thus we feel it is safe to conclude that explicit rule learning, if that took place, was not the dominant process driving choice behavior for the risky cue.

*3) If the RSTD model is validated, it would be useful to also present results relating to the learning rates directly, not just the difference index, as authors attempt an interpretation in terms of LTP/and or LTD.*

We agree with the reviewer that these results should be incorporated into the main text. There were no significant differences between the groups in each of the learning rates on its own, but rather only in the asymmetry index (which quantifies their relative contribution to learning, scaled by the overall rate of learning).

Upon reflection, and based on this comment, we realized that there is another way to specify the RSTD model that would more intuitively illustrate this main modeling result – that the differences between the groups were due to learning asymmetry and not overall levels of learning. The RSTD model can be rewritten with a single learning rate (that determines the rate at which new outcomes change the current value, which is equivalent to the rate in which the impact of previous outcomes is phased out) and a second asymmetry parameter. In fact, this is how the model was originally specified by Mihatsch & Neuneier (2002):Vnew(cue)= Vold(cue)+η∙δ∙(1±κ)

where δ is the prediction error, η is the (single) learning rate and κ is an asymmetry parameter that is added or subtracted depending on whether the prediction error is positive or negative, respectively. In this formulation, the learning rates are not different between the groups, but the κ parameter is. This is the same result that we previously reported, as in this specification of the model η+= η(1−κ) and η−= η(1+κ), giving κ= η−− η+η−+ η+ which was our asymmetry index that did differ between the groups, and η= η−+ η+2, the average of our two learning rates (that did not differ between the groups). Given that this framing seems clearer, we have now rewritten the model in the manuscript in this form (calling it a learning asymmetry model). As requested, we also provide the mean values of each of the fit parameters for both groups, together with their statistics:

"We found significant differences between the groups in the learning asymmetry parameter (DYT -0.05 ± 0.27, CTL -0.34 ± 0.27, Mann-Whitney z=-2.51, df=24, P<0.05), but no differences in the other two parameters (learning rate DYT1 0.25 ± 0.19, CTL 0.14 ± 0.11, Mann-Whitney z=1.33, df=24, P=0.18; inverse temperature DYT 0.68 ± 0.37, CTL 0.93 ± 0.47, Mann-Whitney z=-1.18, df=24, P=0.23)".

4) It would also be important to know whether previous rewards have differential effects depending on whether the trial was a free or forced choice trial (e.g. Figure 3). Recent work (Cockburn et al) has shown that value learning, especially from positive prediction errors, differed between these conditions, and if patients and controls had baseline differences in risk taking for non-learning reasons, this would lead patients to experience more free-choice risky trials than controls, potentially biasing learning. This might be important to rule out.

Given that patients chose the risky cue more often than did controls, and given that there was a fixed number of forced trials that did not depend on group or choices, by necessity DYT group participants chose the risky cue more often in the free choice trials. Therefore, they experienced more learning from the risky cue in choice trials than did the control group. However, it is not clear to us how this could interact with a non-learning account of the results.

In any case, following the Reviewer's comment, and due to the relevance of recent work by Cockburn et al. (2014) to studies such as ours, we examined separately the probability of choosing the risky cue over the sure cue following wins or losses, after either forced or choice trials. Our analysis revealed that choices were significantly dependent upon the previous outcome of the risky cue (P<0.01, F=7.45, *df*=1 for main effect of win versus loss; 3-way ANOVA with factors outcome, choice and group) but not upon its context (P=0.38, F=0.93, *df*=1 for main effect of forced vs. choice trials), as seen in Figure 3—figure supplement 1). We note, however, that similar to Cockburn et al., we indeed observed a numerically smaller effect of the outcome of forced trials (as compared to choice trials) on future choices, as can be seen in the smaller difference between wins and losses in forced as compared to choice conditions (interaction between outcome and choice not significant – P=0.46, F=0.56, *df*=1).

To address this important issue in the manuscript, the caption to this Supplemental figure reads: “Recent work on similar reinforcement learning tasks has shown that choice trials and forced trials may exert different effects on learning (Cockburn et al., 2014). […] Similar to Cockburn et al. (2014), we did observe a numerically smaller effect of the outcome of forced trials (as compared to choice trials) on future choices, however this was not significant (interaction between outcome and choice P=0.46, F=0.56, *df*=1). *P* values in the figure reflect paired t-tests.”

5) It would be interesting to report patients' behavior in absolute, not only relative to controls. It appears that they are in average risk neutral, which is not a suboptimal or irrational thing to do in this task. It might be interesting to discuss why their impairment appears to "correct an imbalance" that is seen in healthy controls, contrary to what is usually observed in patient studies.

We thank the reviewer for this comment. We had reported the absolute behavior of patients:

"Overall, the probability of choosing the risky cue was significantly higher among patients with dystonia than among healthy controls (Figure 3, probability of choosing the risky cue over the sure cue DYT 0.44 ± 0.18, CTL 0.25 ± 0.20, Mann-Whitney z=2.33, df=24, P<0.05)."

Following the Reviewer’s suggestion, we also now added the following text to the final paragraph of our Discussion:

"Relative weighting of positive and negative outcomes shapes our risk-sensitivity in tasks that involve learning from experience. […] In any case, these reinforcement-learning manifestations of what has been considered predominantly a motor disease provide support for linking corticostriatal synaptic plasticity and overt trial-and-error learning behavior in humans."

*Reviewer #3:*

*Building on animal work showing that DYT1 dystonia animal models are associated with exaggerated LTP and diminished LTD, the authors outline an experiment testing for learning abnormalities in human DYT1 dystonia patients. The authors hypothesize that DYT1 patients will show a positive learning bias owing to increased LTP and/or reduced LTD. Consistent with their hypothesis, DYT1 patients do indeed show atypical responsivity to outcomes in that they do not exhibit risk aversion as did matched controls.*

The authors correctly emphasize the chasm between synaptic modification and behavior. The application of genetically linked animal models and human disease states in concert with an algorithmic description of a learning mechanism offers an exciting and constructive path forward. However, this depends critically on a shared mechanism between the animal model and the human brain. I cannot speak to the quality of the animal models or the nature of DYT1 dystonia; however, the text leaves some question as to the shared commonality. The authors point to atypical LPT/LTD in animal models, but abnormal dopamine/acetylcholine transmission in humans. Given the manuscript's emphasis on lessening the gap between synapse and behavior, I feel that the manuscript could benefit from more support linking animal and human disorders (perhaps via behavioral patterns in the animal models, mechanisms through which medications operate etc…).

We thank the Reviewer for this comment, and apologize for the apparent contradiction in the text. We agree with the Reviewer that the gap between animal models and human patients is still wide. Increased LTP/LTD ratio in corticostriatal synapses in rodent models of DYT1 dystonia is a persistent finding in a variety of transgenic models. This LTP/LTD ratio is determined by complex interactions between numerous players and currently it is not known what is the role of each of these players (Calabresi et al., 2014). We discuss this issue at length in response number 1 to Reviewer 1 – we would like to refer the Reviewer to that discussion.

In addition, and in support of our assumption that the findings regarding the LTP/LTD (im)balance are relevant to humans, the LTP/LTD impairment is restored by anticholinergic agents (such as trihexyphenidyl) that are routinely given to patients with DYT1 dystonia (Martella et al., 2009). We agree with the Reviewer that further studies are needed in order to show this relationship conclusively, as so far there is only circumstantial evidence.

Finally, to our knowledge, risk-sensitivity in DYT1 dystonia animal models has never been tested. This is an excellent idea for a future study that we hope our findings will help spur.

Of greater concern is whether these results truly inculpate a reinforcement learning mechanism. As outlined out by the authors, patients exhibited a strong propensity to pick the risky option again following a win but avoided it following a loss, which is argued to demonstrate outcome sensitivity. But, this response pattern does not necessitate a reinforcement learning strategy. These data also appear to be consistent with a win-stay/lose-shift strategy (WS/LS). Given the 50% chance of reward on the risky stimulus, a WS/LS strategy is consistent with the reported lack of risk aversion. Furthermore, the model generated response pattern illustrated in Figure 1 (bottom panel), suggests that a reinforcement learning strategy becomes increasingly risk averse. There does appear to be some trend of increasing risk aversion in controls, but not so for the patient group. Given that there are no clear response patterns that demonstrate a reinforcement learning strategy (i.e. there are no learning curves etc…), I feel that the results would be more interpretable if a WS/LS model was also included in the analysis and Discussion.

We thank the Reviewer for this important comment, which was also raised by Reviewer 2. To avoid repetition, we refer the Reviewer to our answer to Reviewer 2’s point 2C above, where we test the WSLS model explicitly in both a model-based and two model-free analyses.

Regarding the simulation in Figure 1. We used for this simulation the average values for each of the groups in order to qualitatively demonstrate our hypothesis – higher LTP/LTD ratio is translated to increased implicit value of the risky cue and observed as risky behavior. This point is now clarified in the figure legend of our revised manuscript.

In general, the interaction between choice and valuation in reinforcement learning indeed leads to risk aversion, with more risk aversion as the learning rate increases (see Niv et al., 2002 for a mathematical proof of this claim). Intuitively, if the risky option at some point has a lower value than the sure option, the risky option will be chosen less often and thus its value will not be updated, leaving it lower than the sure option (whose value does not change) for a disproportionately long period of time. Therefore, the results of the model simulation and the control group indeed conform to what would be expected from a reinforcement learner. This finding, on its own, however, cannot fully explain the risk sensitivity of even control participants, as they are not always risk averse. Indeed, in Niv et al. (2012), we compared this account for risk sensitivity to the alternative RSTD model, and showed that the latter model better explains choice behavior. The asymmetric learning in the RSTD model allows one to “correct” the imbalance inherent in reinforcement learning, and to achieve risk neutrality or even risk seeking behavior by learning more from positive prediction errors than from negative prediction errors. This is exactly what we show here for the DYT group, and it indeed is mirrored in the fit parameters of the model. Therefore, we don’t agree with the Reviewer that our reinforcement-learning simulations suggest results that are at odds with the behavior of our participants.

Could observed effects in the patient group be driven by medication withdrawal? The authors offer helpful discussion of medication half-life, but this only serves to demonstrate that effects are probably not driven by the medication itself.

We thank the Reviewer for raising this issue. Indeed, withdrawal symptoms were described with both baclofen (Terrence and Fromm, 1981) and trihexyphenidyl (Mclnnis and Petursson, 1985) but typically appear following medication withdrawal for at least a few days or weeks. Patients with dystonia do not typically experience symptoms of withdrawal before their next dose of medication (as in our study) or even when they skip a scheduled dose. This stands in contrast with the 'early wearing off' phenomenon observed in patients with advanced Parkinson's disease. Moreover, risk-taking behavior (Figure 3—figure supplement 3) was also observed in these patient who were not taking any medications, although this sub-group was very small.

We agree with the Reviewer that such an explanation cannot be absolutely ruled out and are aware for this methodological limitation imposed by our choice to work with human subjects. Still, based on the arguments above, we believe it is not a plausible explanation for our observation. In any case, we now also performed a linear regression to verify that our results hold even when accounting for medication:

“Risky behavior was not significantly affected by sex (Figure 3—figure supplement 3) or the patient's regime of regular medication (Figure 3—figure supplement 4), and the relationship between risk taking and symptom severity held even when controlling for these factors (p<0.05 for symptom severity when regressing risk taking on symptom severity, age and either of the two medications; including both medications in the model lost significance for symptom severity, likely due to the large number of degrees of freedom for such a small sample size; age and medication did not achieve significance in any of the regressions).”

What were the model parameter ranges explored while fitting the data? Within subject variance (Figure 3) seem to indicate considerably more variance in the control group (though I understand this to be a subset of the dataset), so I am a bit surprised by the apparent lack of effect in the inverse temperature parameter.

We thank the reviewer for this important comment. The explored ranges of model parameters were 0-1 for both positive and negative learning rates (now -10 to 10 for the learning asymmetry parameter) and 0-30 for the inverse-temperature parameter. This information is now mentioned in our revised manuscript (Materials and methods; Modeling).

We chose to demonstrate changes in risk-taking policy (due to previous exposure and certain randomness captured by our model) by showing behavior of only a few randomly chosen subjects. This was done for the sake of visual clarity and does not reflect the entire group. We note that in this figure, the within-subject variance reflects the combined effects of exposure to rewards in previous choices (these were not equated trial-by-trial across participants), learning rates (higher learning rates lead to rapidly changing policy) and noisy behavior represented by the inverse temperature parameter.

Following the Reviewer’s comment, and in order to directly test the within-subject variability in risk-taking (probability of choosing risky cue) across trial bins (shown in Figure 3) we calculated, for each individual separately, the standard deviation across bins. This value was 0.13 ± 0.07 for both groups.